# Neuroformer: Multimodal and Multitask Generative Pretraining for Brain Data

**Antonis Antoniades,**\* **Yiyi Yu, Joseph Canzano, William Wang, Spencer LaVere Smith**
University of California, Santa Barbara

## Abstract

State-of-the-art systems neuroscience experiments yield large-scale multimodal data, and these data sets require new tools for analysis. Inspired by the success of large pretrained models in vision and language domains, we reframe the analysis of large-scale, cellular-resolution neuronal spiking data into an autoregressive spatiotemporal generation problem. Neuroformer is a multimodal, multitask generative pretrained transformer (GPT) model that is specifically designed to handle the intricacies of data in systems neuroscience. It scales linearly with feature size, can process an arbitrary number of modalities, and is adaptable to downstream tasks, such as predicting behavior. We first trained Neuroformer on simulated datasets, and found that it both accurately predicted simulated neuronal circuit activity, and also intrinsically inferred the underlying neural circuit connectivity, including direction. When pretrained to decode neural responses, the model predicted the behavior of a mouse with only few-shot fine-tuning, suggesting that the model begins learning how to do so directly from the neural representations themselves, without any explicit supervision. We used an ablation study to show that joint training on neuronal responses and behavior boosted performance, highlighting the model's ability to associate behavioral and neural representations in an unsupervised manner. These findings show that Neuroformer can analyze neural datasets and their emergent properties, informing the development of models and hypotheses associated with the brain.

## 1 Introduction

Systems neuroscience experiments are growing in complexity with a cascade of technical advances. In addition to recording neuronal activity from hundreds to thousands of neurons in parallel from multiple brain areas (Yu et al., 2021; Stirman et al., 2016; de Vries et al., 2020a;b), experimenters are also simultaneously acquiring behavioral data including eye positions and body movements (Stringer et al., 2019; Steinmetz et al., 2019). Neuroscientists seek models for these rich datasets to obtain insights into how neural circuitry is involved with behavior. However, complex datasets and experimental designs present challenges for model development, particularly in incorporating the various modalities, and subsequently creating models for a host of different applications.

In recent years, deep neural networks (DNNs) have demonstrated their potential in modeling neural activity and circuitry (Yamins et al., 2014; Kubilius et al., 2018; Richards et al., 2019). Brain-inspired DNNs have exhibited responses similar to visually evoked neural responses in visual cortical areas (Yamins et al., 2014; Kindel et al., 2019; Mineault et al., 2021; Yamins & DiCarlo, 2016). These models were trained in a goal-driven approach, such as the classification of images. Then intermediate representations in the model were compared to observed neuronal activities in biological brains. In other work, autoencoder-based latent variable models have been applied to the analysis of neuronal activities, with the assumption that the single-trial spiking activity of a behavior task depends on underlying dynamics (Sussillo et al., 2016; Ye & Pandarinath, 2021; Zhao & Park, 2017; Schneider et al., 2022). These approaches can provide insight into principles of neural circuitry, but often entail inductive biases that are not always ideal. General purpose, multimodal models can provide a complementary approach.

---

\*Contact: antonis@ucsb.edu, Code: https://github.com/a-antoniades/Neuroformer

To develop models for correlating neuron activities, sensory input, and behavior, we sought a powerful multimodal architecture and training objective. The transformer architecture has demonstrated its flexibility in modeling data from various domains, finding widespread adoption in applications including vision, language, and sound (Liu et al., 2021a; Brown et al., 2020a; Huang et al., 2018; Radford et al., 2021; Gan et al., 2020; Li et al., 2021). In particular, transformer-based language-vision models exhibit high performance for learning multimodal representations for text and images, enabling text-to-image generation (Ramesh et al., 2021; Radford et al., 2021). Key to this is that a) the transformer is scalable; i.e it does not suffer from the vanishing gradient problem (Pascanu et al., 2012) b) it makes few assumptions about the input data (minimal inductive bias) (Jaegle et al., 2021b;a) and c) it can contextualize information within long-duration multimodal contexts. Here we present a transformer-based model for multimodal datasets in systems neuroscience, including neuronal activity, stimuli, and behavior. We developed a generative model which firstly aligns and subsequently fuses neuronal activities, input signals (e.g., visual stimuli) and behavior.

**Our key contributions are as follows: a)** Introduce an approach that reframes the population response inference problem as a causally-masked, spatiotemporal autoregressive generation problem. **b)** Show how to employ a generative pretraining paradigm for neural data. **c)** Investigate the potential of multimodal, multitask, generative transformer models to analyze and parse neuroscience data, and obtain insights.

## 2 RELATED WORK

**Representational and Functional Similarities between DNNs and Mammalian Brains** There are similarities between the hierarchical representations of convolutional DNNs and the visual and inferior-temporal cortices (Yamins et al., 2014; Kubilius et al., 2018; Kindel et al., 2019; Mineault et al., 2021; Yamins & DiCarlo, 2016). More recently, parallel-path networks trained using a contrastive-predictive objective were shown to separate into pathways that mirrored the functional specializations of the dorsal and ventral streams in mouse visual cortex (Bakhtiari et al., 2021). Similarly, there are parallels between human speech and language models (Caucheteux & King, 2022; Millet et al., 2022). Lastly, reports have shown the plausibility of transformers replicating specific neural functions and circuits (Whittington et al., 2021; Bricken & Pehlevan, 2022). In our work, establishing a similarity between a DNN and the brain is not our focus. Instead, we are focused on developing an analytical and functional tool. This tool, Neuroformer, can exhibit similarities in representational and functional characteristics with biological neural circuitry, and we explore that here.

**Learning Latent Dynamics of Population Activity** Variational auto-encoders have been used to infer latent dynamics embedded in complex, high-dimensional neural data (Sussillo et al., 2016; Zhou & Wei, 2020). Recently, a transformer architecture has been used in place of RNNs for the same application (Ye & Pandarinath, 2021), and another project has trained a latent model using a contrastive objective (Schneider et al., 2022). These empirical findings show that this approach avoids over-fitting and outperforms previous methods. In our work, we integrate these paradigms, by employing a contrastive objective at the front end of a generative model.

**Large Multimodal and Multitask Models** The transformer has proven effective at processing a variety of modalities including vision, languange, and sound (Liu et al., 2021a; Brown et al., 2020a; Huang et al., 2018). Subsequently, the field introduced multimodal models. For example, models for interactions between images and text (Tan & Bansal, 2019; Chen et al., 2019; Lu et al., 2019). Some models attempt to align the modalities with separate feature encoders using contrastive learning approaches (Radford et al., 2021; Gan et al., 2020), and alignment and fusion (Li et al., 2021). We experimentally validate these methods on neural data, by using our own generative and contrastive learning objective. Furthermore, architecture innovations have resulted in models that are agnostic to the input modalities and can be applied to a large variety of tasks (Jaegle et al., 2021b;a). Lastly, it has been recently observed, that large models pretrained on massive datasets of text exhibit emergent properties. This has been observed in Large Language Models (LLMs) as in (Brown et al., 2020b; Wei et al., 2023; Jung et al., 2022; Creswell et al., 2022; Ganguli et al., 2023), but also in scientific domains (Rives et al., 2021; Bakhtiari et al., 2021). We take inspiration from these lines of work to

build a framework for modelling neural data that can efficiently scale to large inputs and an arbitrary number of modalities.

## 3 MODEL

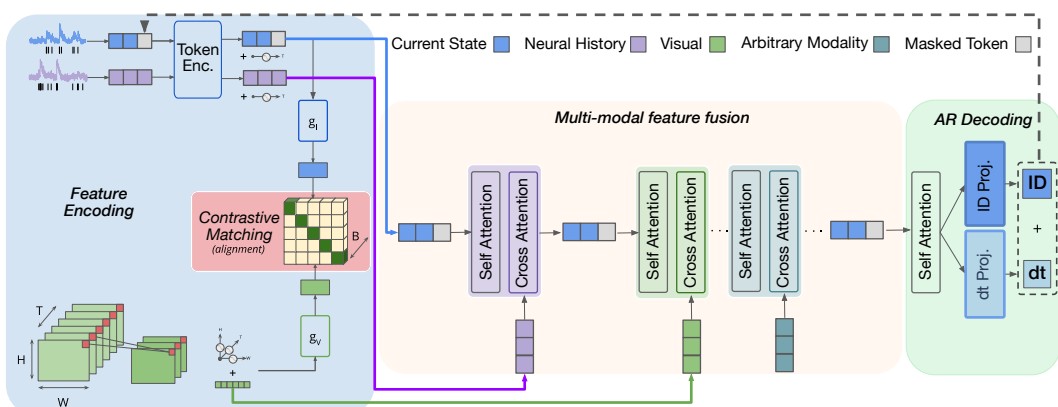

Figure 1: Neuroformer architecture. Inputs undergo contrastive matching to ensure efficient representations for downstream processing. Neural activity, stimuli, and other modalities are added to the Current State using cascading self- and cross-attention modules. Finally, the output of spike predictions (neuron ID and time interval tokens) are read out. During inference time, neuron IDs autoregressively populate the block, and the corresponding predicted time tokens act as additive temporal encodings.

**Workflow** As input, our model takes action potential data, i.e., spike trains, from multiple neurons recorded or imaged simultaneously, and breaks them up into the Current State $I_c$, and past state $I_p$ according to specified temporal context windows $(w_c, w_p)$ (**Fig. 1**, blue and purple, respectively). We do this to discretize them and precisely align the neural representations with features from other modalities. Model inputs, including neural, visual, and other features, are optionally processed by feature encoders. The modalities are first aligned via a multimodal contrastive module (Radford et al., 2021; Chen et al., 2020; Oord et al., 2018) which aims to learn matching representations for the neural features and the features from other modalities. Subsequently, a cross-modal transformer fuses the current neural response $I_c$ with all other features. We then decode our fused representation using a causally-masked transformer decoder. All training is self-supervised and requires no data labelling.

**Feature Backbones** We assign an ID (number) to each neuron in our dataset and model it as a token. At each step, a learnable look-up table projects each neuron spike contained in our *Current* and *Past States* onto an embedding space $E$, resulting in vectors $(T_c, E)$ and $(T_p, E)$, where $T_c, Tp$ are the corresponding state's sequence length (number of spikes + padding). For video frames, we optionally use a single layer of 3D Convolutions (Hara et al., 2017) before splitting them into patches of size ($T_f, H, W, E$), as is standard for visual transformers (**Fig. 1**, Feature Encoding, green) (Dosovitskiy et al., 2020; Liu et al., 2021b; Li et al., 2019). We then feed these patches into the cross-modal layers.

**Multimodal Contrastive Representation Learning** Drawing from recent developments in vision and language representation learning (Radford et al., 2021; Zellers et al., 2021; Li et al., 2021), our approach leverages a contrastive learning objective before fusing our feature sets (**Fig. 1, Contrastive Matching**). Our architecture allows alignment of an arbitrary number of modalities; however, here we focus on the three-modal case. We use Visual features $F_{p,c} \sim (B, T, H, W, E)$, *Current State* neural embeddings $I_c \sim (B, T_c, E)$, and *Behavior* features $A_c \sim (B, T_a, E)$ as our modality representatives, where $B$ = Batch Size and $T$ = sequence length. These are projected to a latent dimension $d$ using three linear layers $\boldsymbol{g}_f, \boldsymbol{g}_i, \boldsymbol{g}_a$. We calculate pairwise similarities between the modalities, yielding three matrices of dimensions $(B_f, B_i), (B_i, B_a), (B_a, B_f)$ where

$B_f = B_i = B_a = B$. The diagonal entries correspond to ground-truth pair similarities. These scores are then softmax-normalized and used to compute the contrastive loss via cross-entropy H, where the probability $\boldsymbol{p}$ should be 0 and 1 for the negative and positive pairs $\boldsymbol{y}$, respectively. We generalize the computations with pair notation (m,n) where m and n are any of the modalities (either F, I, or A):

$$s(m,n) = g_m(m_{p,c})^T g_n(n_c) \qquad\qquad p_k^{mn} = \frac{\exp(s(m, n_k)/\tau)}{\sum_{k=1}^{K} \exp(s(m, n_k)/\tau)} \qquad (1)$$

We then apply these equations to each pair (F,I), (I,A), (A,F), and compute the contrastive loss:

$$L_{vna} = \frac{1}{3}\,\mathbb{E}_{(F,I,A)\in d}[H(\boldsymbol{y}^{fi}(F), \boldsymbol{p}^{fi}(F)) + H(\boldsymbol{y}^{ia}(I), \boldsymbol{p}^{ia}(I)) + H(\boldsymbol{y}^{af}(A), \boldsymbol{p}^{af}(A))] \quad (2)$$

By explicitly imposing alignment, we encourage the model to uncover the representational commonalities between biologically relevant features (Schneider et al., 2022). This is particularly helpful in low-data settings, mitigating the over-fitting that can occur with variable neural signals. We observe consistent performance improvement upon adding the contrastive objective (see **Fig. 6, 10**.)

**Feature Fusion**    Following alignment, the masked *Current State*, $I_c$ is fused with all other features, in our case the visual $F$ and neural *Past State* features, $I_p$ by cascading cross-attention modules. This stage of the model is similar to the (Jaegle et al., 2021b;a), where cross-attention between a smaller latent array $(T_c, E)$, in our case the *Current State* and larger byte arrays $(T_f, sE)$, in our case the *Past State* neural features, is iteratively carried out. This architecture decision has two main advantages: a) It introduces a bottleneck imposed by the smaller $I_c$ array, of length size $T_c$ that upon operating with a larger feature array of length $T_f$, reduces the complexity of attention from $\mathcal{O}(T_f^2)$ to $\mathcal{O}(T_c T_f)$. When $T_c$ is much less than $T_f$ (expressed mathematically as $\lim_{T_c\to 0, T_f\to\infty} T_c/T_f = 0$), the complexity scales linearly with $T_f$. This enables the use of finer-grained features, like large-resolution visual features or in our case raw video frames, which are essential for modeling lower-level visual areas. b) The iterative nature of the algorithm enables us to fuse an arbitrary number of modalities with our *Current State*.

**Causal Spike Modelling**    After it has been fused with all other modalities, the resulting *Current State* is decoded using a causally masked transformer (Brown et al., 2020a). At each prediction step, the model predicts both which neuron will fire, and within which sub-interval $(\boldsymbol{dt(n)} = n \in w_c)$, using two linear projections from the last self-attention layer **(Fig. 1, AR Decoding)** mapping to the corresponding $(\boldsymbol{p}_i, \boldsymbol{p}_{dt})$ label distributions. Notice that two Spikes can occur at the same time, and therefore by predicting *bins* within which neurons fire, we can sequentially predict spikes that occur simultaneously. We optimize for this objective, by minimizing the cross entropy between the predicted and ground truth distributions:

$$L_{ce(I)} = \mathbb{E}_{(I)\sim d}\, H(\boldsymbol{y}_I, \boldsymbol{p}_I) \qquad\qquad L_{ce(dt)} = \mathbb{E}_{(dt)\sim d}\, H(\boldsymbol{y}_{dt}, \boldsymbol{p}_{dt})$$

**Loss Function**    The model is jointly optimized end-to-end using the contrastive objective $L_{vnc}$, and cross-entropy loss for both $I_c$ and $dt_c$, $(L_{ce(I)}, L_{ce(dt)})$. The resulting loss is a weighted average of the three losses:

$$L = (\gamma)L_{vnc} + (\mu)L_{ce(I)} + (1 - \gamma - \mu)L_{ce(dt)} \qquad (3)$$

**Inference - Generating Simulations**    At inference time, the model autoregressively predicts the distributions of the $n$th step, $(I_c(n), dt_c(n))$, outputting an *end-of-state* [EOS] token to indicate it has completed the *Current Target State* $W_c$, in a similar fashion to work in image-text generation (Ramesh et al., 2021), **(Fig. 1, AR Decoding)** . To sample from the respective distributions $(\boldsymbol{p}_I, \boldsymbol{p}_{dt})$ we use nucleus sampling (Holtzman et al., 2019). The predicted *Current Target State* neuron IDs are then incorporated into the *Past State* according to the *Past State Context Window* $w_p$ together with the predicted sub-intervals $dt_c$, which act as additive temporal embeddings **(Fig. 1, dotted arrow from AR Decoding)**.

**Finetuning - Transferring to New Tasks**    A primary motivation for redefining the spike inference problem in a manner compatible with contemporary pretraining paradigms is the ability to utilize a broad spectrum of techniques developed within this framework. To transfer to a new task, whether it be behavioral prediction or any other downstream task, we adopt a straightforward heuristic. First, if necessary, the data distribution is discretized into $n$ classes. Then, a mean-pooled representation from our final latent layer is projected onto the appropriate feature space $p_v \in n$. The loss is calculated using the standard multi-class cross-entropy for classification (Howard & Ruder, 2018; Yosinski et al., 2014; Hinton et al., 2015) of mean-squared error for regression.

**Training**    We trained all Neuroformer models with 8 layers and 8 heads for the decoder and each of the cross-attention modules, with an embedding dimension of 256 and an $80/20$ train/test split. V1+AL and Visnav models equated to a total of $40,000,000$ and $100,000,000$ parameters respectively. We emprically observed an increase in performance up until the specified parameter sizes. We used AdamW optimizer (Loshchilov & Hutter, 2019) in PyTorch. Learning rate was 2e-4, with linear warmup and cosyne decay schedules. Batch size was kept at 224 for pretraining and 32 for fine-tuning.

## 4    RESULTS

To evaluate performance of Neuroformer, we used both artificial data sets, with known ground truth, and real neural data sets, with authentic variability and real-world relevance. We analyzed the performance in terms of inference and features.

### 4.1    DATA

**Simulated Dataset**    We simulated a spiking neural network characterized by directional connectivity with three hub neurons using Brian2 simulator (Stimberg et al., 2019) (Hub-net simulation). The network has $N = 300$ leaky integrate-and-fire neurons with stochastic current. The membrane potential ($V_i$) of each neuron was simulated as a stochastic differential equation:

$$\frac{dV_i}{dt} = \frac{I - V_i}{\tau} + \sigma\sqrt{\frac{2}{\tau}} \cdot x_i \tag{4}$$

The membrane time constant was set to $\tau = 10$ms as a Gaussian random variable with zero mean and 1 standard deviation. We scaled the random variable by $\sigma = 0.5$. The total dataset equated to 1 million tokens.

**Two-photon calcium imaging datasets**    We assessed the performance of Neuroformer using real neural datasets. These consisted of neuronal activity recorded from awake mice, which were either passively viewing visual stimuli (Passive-Stim data), or actively engaged in a visually-guided navigation task (Visnav data). The activity was recorded using two-photon calcium imaging, (Dombeck et al., 2007; Yu et al., 2021; 2022) and neuronal spikes were deconvolved from calcium traces via the suite2p calcium imaging processing pipeline, (Stimberg et al., 2019) complemented by a Bayesian spike inference method (Pnevmatikakis et al., 2013). The Passive-Stim data comprised recordings from 386 neurons in the primary visual cortex and a higher visual area (V1 + AL). These neurons responded reliably to drifting grating stimuli and a naturalistic movie. Two Visnav datasets were derived from recordings in the lateral visual cortex (2022 neurons) or L2/3 medial (1905 neurons), spanning V1 and multiple higher visual areas. During these recordings, the mice were engaged in a visually-guided navigation task within a virtual environment, with their movements tracked by a ball system (Harvey et al., 2009) that controlled the first-person view within the virtual environment. The Visnav neurons conveyed information about both the visual input and the speed of the animal's movements. The resulting dataset sizes equated to $80 \times 10^3$ , $800 \times 10^3$ and $1 \times 10^6$ tokens.

### 4.2    ATTENTION REVEALS THE FUNCTIONAL CONNECTIVITY OF SIMULATED NEURONAL CIRCUITS

To test our hypothesis that attention can provide insights into neural circuitry, we first trained the Neuroformer on a simulated dataset with known ground-truth connectivity. This dataset is unimodal,

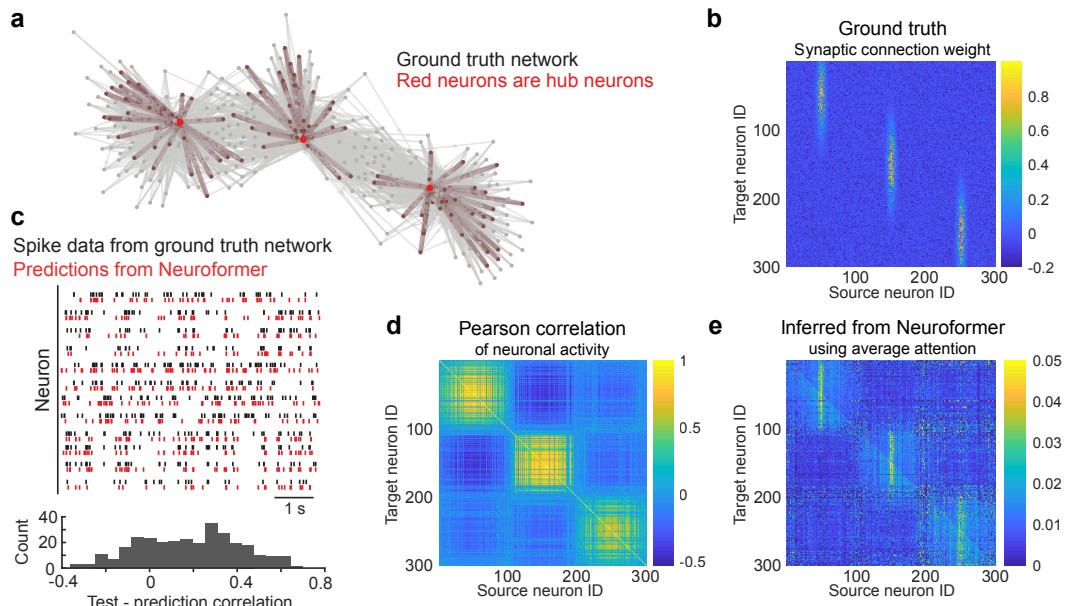

Figure 2: Validating Neuroformer with ground truth data. (**a**) A model spiking neural network was generated to provide ground truth for validation. To provide a salient network feature, three "hub" neurons are strongly connected to many other neurons. (**b**) The ground truth connectivity matrix provides a reference for the network. (**c**) Raster plots (top) for neurons show that Neuroformer provides spike predictions (red) that closely match the spiking of the ground truth network (black). Correlations between the predicted and ground truth spike trains were generally positive, and ¿0.1 (bottom). (**d**) A simple Pearson correlation analysis reveals three subnetworks, but not the hub neurons, because of the lack of directionality or causality. (**e**) Attention mechanisms in Neuroformer infer causality and thus reveal the hub neurons.

so we simply omitted the visual cross-attention module, essentially reducing to a vanilla GPT architecture. After training, we ran masked inference on the data, aggregating the attention values that are assigned to each neuron, and counting how many times neurons attend to each other during this process. This results to two square matrices $N_a$, $Nz$ of size $(N, N)$, where $N$ is equal to our neuronal population's size $(300)$. By dividing $(N_a)$ by $(Nz)$ we compute the average attention $a_{av.}(i, i)$ that neurons assign to one another. Our results recovered the hub connectivity structure of the ground truth neuronal circuit ( $I \in 50, 150, 200$ ) using only the spiking activity (**Fig. 2**). This attention analysis identified the neurons that were driving circuit activity. Compared to a simple correlation matrix, the attention analysis matrix provides directional, potentially causal, information.

### 4.3 REALISTIC AUTOREGRESSIVE SIMULATION OF MULTIMODAL NEURONAL DATASET

To validate Neuroformer as a viable analysis tool for neuronal population datasets we tested its ability to model real neuronal data. We trained Neuroformer to predict visually evoked neuronal activity in a population of 386 neurons, recorded using two-photon calcium imaging in a mouse. The neurons were recorded simultaneously, and spanned two visual cortical areas: primary visual cortex (V1) and the anterolateral (AL) higher visual area. After training, Neuroformer is able to autoregressively generate whole-trial simulations (96 seconds) of the neuronal population that closely resemble held-out trials (**Fig. 3**).

#### 4.3.1 NEUROFORMER OUTPERFORMS GLM

The Generalized Linear Model (GLM) (Truccolo et al., 2005; Pillow et al., 2008b) is a versatile and widely used statistical modeling framework that generalizes linear regression to accommodate non-normal response variables and non-linear relationships. It has been widely used by neuroscientists as a performant tool for stimulus-response population decoding. Although useful, it has

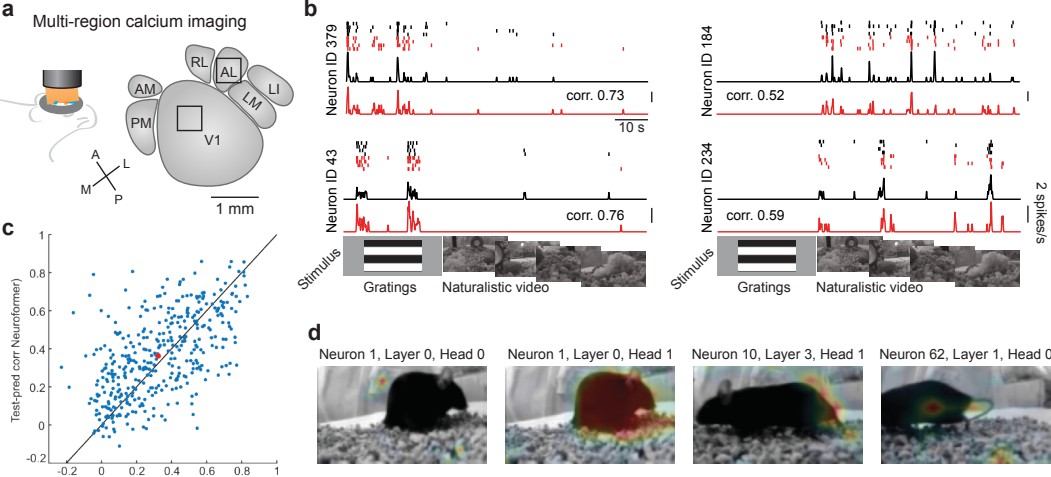

Figure 3: Demonstrating utility of Neuroformer with real data. (**a**) We measured neural activity in two visual cortical areas, V1 and AL, in mice using large field-of-view two-photon calcium imaging. After screening for reliable neuronal responses to stimuli, 386 neurons were included for analysis. (**b**) Neuroformer modeled the spiking data accurately, for responses to two classes of visual stimuli: gratings and naturalistic videos (black ground-truth, red generated). (**c**) Neuroformer vs. GLM population response prediction comparison. Our model's predictions where more correlated with the ground-truth (t-test, $p = 0.0196$) (**d**) Attention provided data-driven maps of the parts of stimuli that had statistical dependencies with neuronal responses.

some significant disadvantages, in that it requires heavy inductive bias (linear kernel, non-linearity, Poisson spiking). This limits both its generality and flexibility in simultaneously integrating many modalities. We compared the predicted, trial-averaged responses from a GLM to Neuroformer on the V1+AL dataset and found that our model's predictions where more closely correlated with the ground-truth (**Fig. 3c**). Our results revealed that our model could better capture the variability across the diverse set of stimuli in the dataset. We empirically observed that a single GLM could perform well on either the diffraction gratings or naturalistic movies, but not both at the same time.

### 4.3.2 BIOLOGICALLY RELEVANT FEATURES EMERGE FROM STIMULUS-RESPONSE CROSS-ATTENTION

Again, we turned to the attention parameters in the Neuroformer, as we had for the hub neuron analysis. We plotted the spatial locations on visual stimulus frames where cross-attention was allocated, i.e., a sort of attention map. These maps exhibited localized features and structure. These maps are related to the concept of receptive fields, but are specific to the full context of ongoing visual stimuli and neuronal activity. Thus, they can reveal novel relationships and insights into potentially causal relationships among neuronal activity and stimuli (**Fig. 3**).

### 4.3.3 HIGHLY ACCURATE AND FEW-SHOT PREDICTION OF MOUSE BEHAVIOR

In this section, we introduce the notion of finetuning (Howard & Ruder, 2018) for neuronal data. By reframing the neuronal modelling problem to be compatible with modern NLP approaches, we can leverage similar techniques to solve tasks that can benefit from neuronal pretraining. One such task is predicting the behavior of a mouse from its neuronal responses. This is a common thing to do in neuroscience analysis, where researchers want to investigate the relationship between neuronal activity and behavior. Furthermore, such paradigms could be useful in brain-computer interface applications. For this analysis, we used neural and behavior data from a mouse running in a virtual reality setup. Neuroformer can support both classification with a mutli-class cross-entropy objective or regression using mean-squared error, which can be chosen according to the nature of the task. Here we report results using regression (**Fig. 4, Table 1**) and have included classification and regression inference results for all models in the appendix (**Figs 11, 12**). Similarly to how multi-

task models in the field of vision and language work, finetuning a model for a new task involves projecting the features of the last layer onto the new feature space. (Howard & Ruder, 2018)

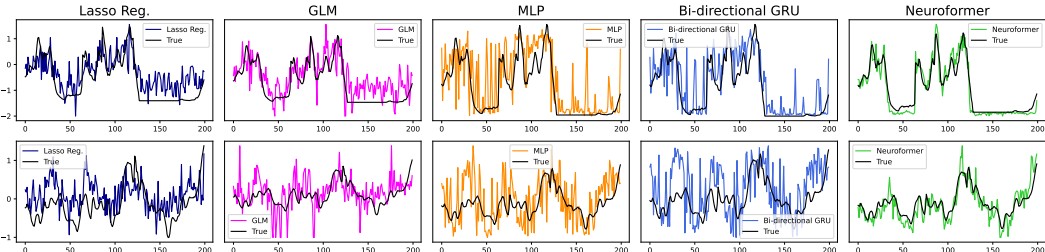

Figure 4: Raw Speed vs. Predicted for all methods for medial (top) and lateral (bottom).

Despite the inherent stochasticity in the speed of the mouse, the predictions of our model when trained on the full dataset are highly correlated with the held-out ground-truth. Additionally, Neuroformer significantly outperformed conventional approaches, achieving pearson $r = 0.97$, with the held-out ground truth for the respective lateral and medial datasets **(Table 1)**. To determine the effectiveness of the masked neuronal prediction pretraining we compared the performance of pretrained vs. randomly initialized models on 10% and 1% of the finetuning data. The pretrained

Table 1: Behavior Prediction

| Model | Pearson Corr (r) | |
| --- | --- | --- |
| | **Lateral** | **Medial** |
| Lasso Regression | 0.62 | 0.73 |
| GLM | 0.69 | 0.81 |
| MLP | 0.83 | 0.85 |
| Bidirectional GRU | 0.83 | 0.88 |
| Neuroformer | 0.97 | 0.97 |

few-shot models quickly approached the performance of the model trained on the full dataset **(Fig. 5b,c)**. Crucially, we observed that the pretrained model that was finetuned on 1% of the data outperformed the randomly-initialized model that was finetuned on 10% of the data. This suggests that pretraining enables the model to begin learning representations that go beyond the optimization objective as has been observed in generative models within natural language generation. (Radford et al., 2019; Sanh et al., 2022)

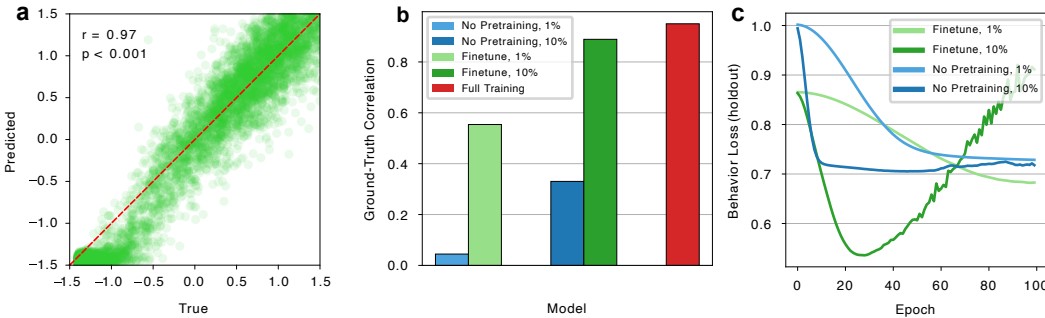

Figure 5: **(a)** Velocity prediction vs. Ground Truth. Neuroformer's predictions were highly correlated (pearson $r_{1\%,full} = 0.97$) with the actual speed of the mouse. **(b)** Pretrained neuronal models exhibit few-shot learning of mouse speed. The pretrained model finetuned using 1% of speed data was able to significantly outperform a non-pretrained equivalent trained on 10% of the data. (pearson $r_{1\%,ft} = 0.51$ vs. $r_{10\%,nopre} = 0.33$ respectively) **(c)** Behavior loss. The pretrained models (green) were able to converge faster and to a lower testing loss compared to the non-pretrained equivalents.

## 5 ABLATIONS

To explore the impact of each component of Neuroformer, we implemented a range of model versions. Starting from just decoding from the *Current State*, we progressively incorporated *Past State*,

*Video*, *Behavioral modalities*, and *Alignment* using contrastive learning. With each addition, we noted an enhancement in the model's ability to generate realistic neuronal responses **(Fig. 6, 10)**.

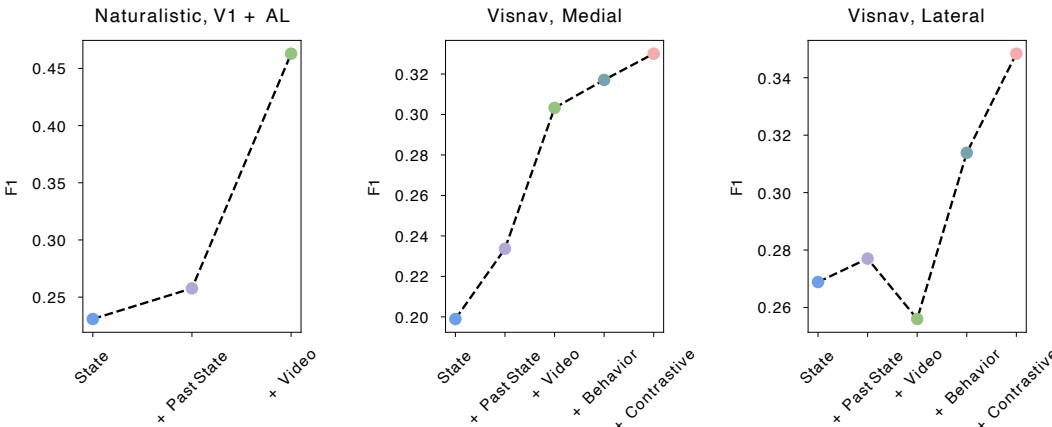

Figure 6: Model component effect on performance. Each of the components (*Current State, Past State, Video, Behavior, and Contrastive*) improve the predictive capability of the model across datasets and brain regions. Points color-coded according to model architecture **(see Fig. 1)**.

Of particular interest, was the Lateral dataset where the behavioral modality we incorporated for the ablation study was eye position (see **Fig. 13** for predictions). We hypothesized that by learning to extrapolate from the eye position to the visual field of the mouse, the model could improve its performance, which is what we observed. While more investigation needs to go into understanding exactly how Neuroformer incorporated eye position within this context, this result highlights the potential of our approach.

By observing the fluctuation in performance as components within the model are selectively incorporated, scientists can examine hypotheses about the relevance of various input modalities or phenomenological behaviors to different brain regions. Ultimately, the development of modular and performant foundation models of the brain will create new avenues for the scientific exploration of the principles that underpin neuronal function and intelligence.

## 6 CONCLUSION

We have developed a new method to enable generative pretraining of multimodal neural data, and have demonstrated its utility accross four novel neural datasets with diverse structure. Our initial observations showed that pretraining not only adopted representations that mirrored the causal patterns found in neuronal circuitry, but was also able to effectively enhance performance in downstream applications, enabling rapid few-shot adaptation to behavior-related tasks.

Our model outperformed conventional approaches, while in comparison incorporating minimal inductive biases and retaining an identical architecture across datasets and experiments. Overall, the model was applied to 4 modalities (neural data, video, speed, eye position), and 3 decoding tasks (neuron ID, neuron dt, speed). The architecture enables easy expansion of the model to more modalities and tasks, making it compatible with almost any type of systems neuroscience dataset and decoding objective.

Concluding, the Neuroformer enables systems neuroscience to leverage the full suite of tools in generative, multitask and multi-modal transformers. It paves the way for scaling training of neuroscience models and potentially connecting them with Large Language Models. This advancement could foster models with new emergent capabilities, benefiting not just neural data analysis but also various other applications. Our work sets the stage for promising research within this domain. [1]

---

[1]For a more thorough discussion of limitations, please see Appendix A.

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

SUPPLEMENTARY MATERIALS

## A  LIMITATIONS

In preliminary experiments, the Neuroformer was trained on simulated data, revealing potential in identifying hub network connectivity. However, its reliability varied across datasets. For instance, in a multi-channel simulation, neuron attention sometimes accurately identified the correct channel, but at other times was noisy. In another test with neurons responding to a naturalistic video via 3D gabor filters, the method's reliability in uncovering receptive fields was inconsistent. We think this technique could benefit from more advanced methods for calculating the attention like Grad-CAM.

Our model, the first of its kind using an auto-regressive decoder-based generative pre-training approach, contrasts with encoder-based methods like LFADS or CEBRA. While these methods explicitly exploit the intrinsic dimensionality of neuronal populations, the Neuroformer, inspired by the GPT objective, takes a different approach. Out attempts to use our data with representation learning approaches yielded poor results (see **Fig. 10**). These methods are more typically applied to motor control tasks for BCI applications. Direct comparisons with methods like LFADS are needed, as they might currently offer better representation learning due to explicit low-dimensional constraints. In contrast, our method is more akin to an in-silico model of the brain.

While transformers, like the Neuroformer, offer advantages like being modality-agnostic and contextualizing information over extended sequences, traditional methods like the GLM and CNNs have their merits. They are grounded in neuroscientific assumptions, offering interpretability. Although the Neuroformer showed promise in decoding tasks, the reasons for its performance advantages remain unclear. Factors like overparameterization, ability to model non-linearities, and effective representation learning might contribute.

Our aim with the Neuroformer was two-fold: 1) to introduce a tool for neuroscience to flexibly incorporate data from increasingly diverse and complex experimental settings and 2) to explore generative pre-training for such data. The true potential of this method might be realized with increased scale, given the vast data available in systems neuroscience. The Neuroformer marks a step towards this, but challenges persist. We're keen on scaling these methods and finding synergies between large language and neuroscience models.

## B  ETHICAL STATEMENT

This project involves data from mouse experiments. Our AAALAC (Association for Assessment and Accreditation of Laboratory Animal Care)-accredited institution is committed to the humane care and use of animals in research and education. Our Institutional Animal Care and Use Committee (IACUC) oversees animal care and use, facilities, and procedures. The attending veterinarian and IACUC ensure that research activities involving mice meet the ethical and legal requirements as set by relevant policies and guidelines.

## C EXPERIMENTAL SETUP

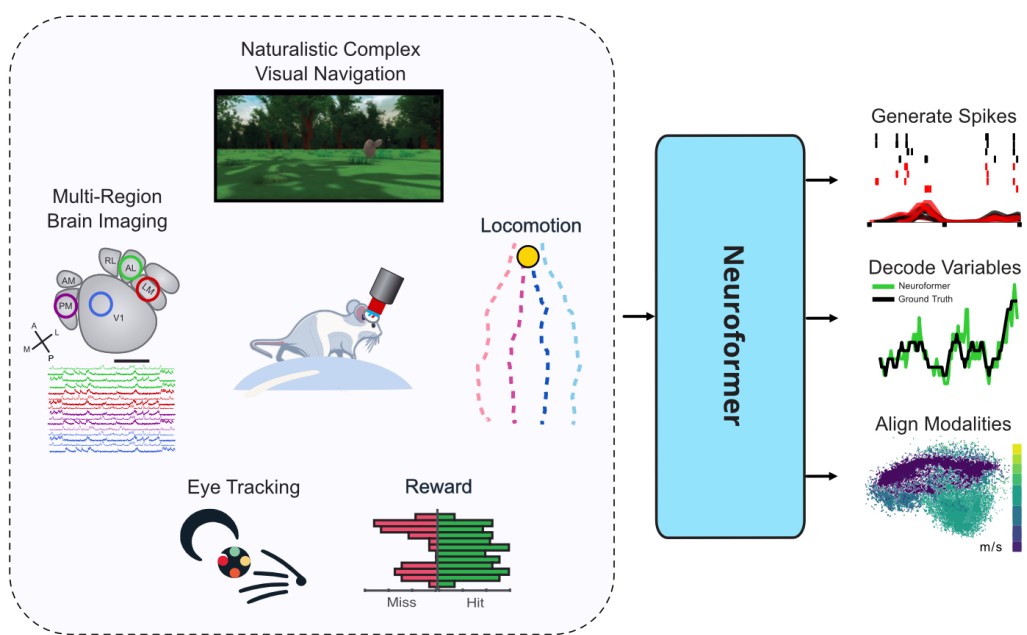

Figure 7: Neuroformer can incorporate a diverse range of modalities that are recorded in large-scale systems neuroscience experiments. Here, we show some of the modalities collected in the visual navigation experiment used to train our model.

# D HYPERPARAMETERS

## D.1 NEUROFORMER

| Hyperparameter | Symbol | V1+AL | Visnav |
|---|---|---|---|
| Video resolution (s) | $dt_{\text{frames}}$ | 0.05 | 0.05 |
| dt token resolution (s) | $dt$ | 0.01 | 0.005 |
| Behavior feature resolution (s) | $dt_{\text{behavior}}$ | - | 0.05 |
| Dropout (p) | $p$ | 0.35 | 0.35 |
| Behavior Dropout (p) | $p_{\text{behavior}}$ | - | 0.45 |
| Contrastive Learning Temperature | $\tau_{\text{clip}}$ | 0.25 | 0.25 |
| Contrastive Learning Latent Dimension (n) | $E_{\text{clip}}$ | 1024 | 1024 |
| Embedding Dimension (n) | $E$ | 512 | 512 |
| Window current state (s) | $w_c$ | 0.05 | 0.05 |
| Window past state (s) | $w_p$ | 0.25 | 0.15 |
| Window frames (s) | $w_{\text{frames}}$ | 0.3 | 0.2 |
| Window behavior (s) * | $w_{\text{behavior}}$ | - | 0.05 |
| 3D Convolution Kernel Size | $c_{\text{kernel}}$ | (6, 8, 8) | (4, 5, 5) |
| 3D Convolution Stride Size | $c_{\text{stride}}$ | (6, 8, 8) | (2, 5, 5) |
| Number of state layers (n) | $n_{\text{state\_layers}}$ | 8 | 8 |
| Number of state stimulus layers (n) | $n_{\text{state\_stimulus\_layers}}$ | 8 | 8 |
| Number of self-attention layers (n) | $n_{\text{self\_attention\_layers}}$ | 8 | 8 |
| Number of behavior layers (n) | $n_{\text{behavior\_layers}}$ | - | 8 |
| Number of heads (n) | $n_{\text{head}}$ | 8 | 8 |

Table 2: Hyperparameters used for training models on V1+AL and Visnav datasets. The units are added in the Hyperparameter column where applicable: s for seconds, n for number (count), and probability for dropout rates.

# E COMPARISON MODEL DETAILS

## E.1 POPULATION RESPONSE

### E.1.1 GLM

A generalized linear model (GLM) for neuron data was adapted from (Pillow et al., 2008a). The instant firing rate ($r_i(t)$) of individual neurons was modeled by a linear-nonlinear cascade, including linear stimulus components ($k * x$) and spike history components ($h * y$) followed by an exponential nonlinearity. $\mu$ represents the baseline firing rate of the neuron. The stimulus kernel ($k$) and history kernel ($h$) are modeled by cosine bases. The parameters are optimized by minimizing the log-likelihood for spike prediction.

$$r_i(t) = exp(k * x + h * y + \mu) \tag{5}$$

## E.2 BEHAVIOR DECODING

### E.2.1 MLP

The MLP used for the behavior comparisons was comprised of 5 stacked MLP layers, with a hidden state of ($h = 1024$), GeLU non-linearity, dropout $p = 0.01$, layer-normalization between each of the MLPs. No improvement was observed upon adding more layers.

### E.2.2 GRU

The GRU model used for the behavior comparisons had the following architecture: 5 Bi-directional GRU layers with dropout $p = 0.01$, layer-normalization, and tapering $t=0.5$ for each consecutive hidden state. Tapering allowed us to stack more GRU layers without undergoing collapse. More layers resulted in a decrease in performance and eventual collapse. The first hidden state of the

model was initialized as the neural *past state* as defined in the paper, allowing the model to integrate past information. We employed Neuroformer-style tokenization which shortened the sequence of spikes and resulted in better performance.

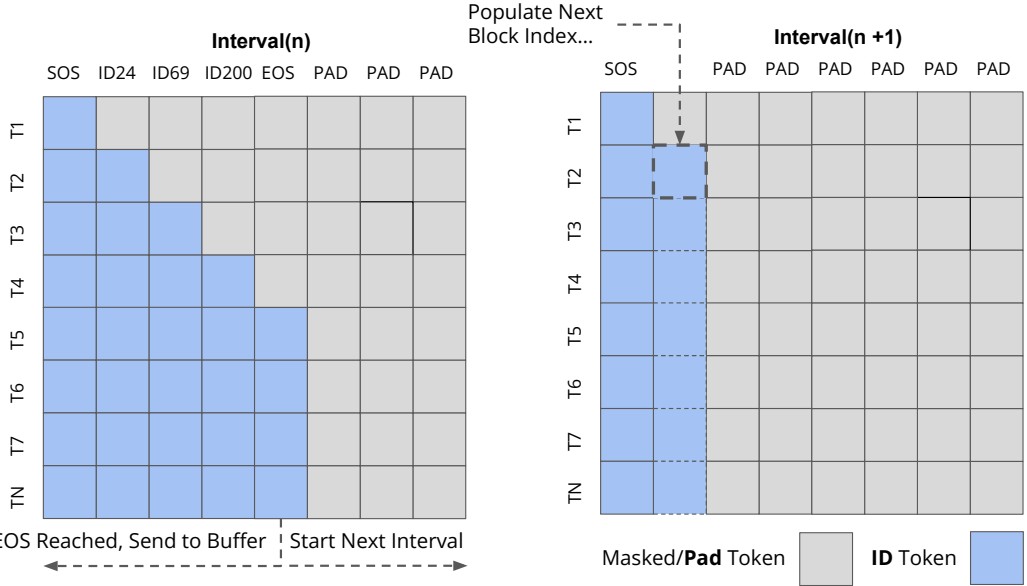

Figure 8: *Current State* causal block masking and padding. During training, the block is populated with all *Current State* spikes, and causally masked. To retain the same length of block, (and enable batched training) we right-pad the remaining sequence, and zero-out the loss corresponding from the `PAD` positions. During Inference, a new *Current State* is initiated, and all corresponding modalities (e.g., video frames) are passed into the model. The block is autoregressively populated starting from the `SOS` position with `ID` tokens, until `EOS` is reached. The predicted spikes are then fed into the buffer, so they can be incorporated into the *Past State*.

## F  BLOCK-SPARSE CAUSAL MASKING

To precisely align the corresponding neural features with all other modalities, we broke our state features into a *Current State* $w_c$ and a *Past State* $w_p$ according to specified temporal windows (see **Table 2**).

Because of this, we cannot simply model the neural features as a continuous string of spikes (or the equivalent language tokens) as in the case of a standard GPT model (Radford et al., 2019). The unique challenge of arranging our data in this way is that each of these intervals contains a different number of spikes. Our situation is similar to (Ramesh et al., 2021), where during inference time, the block is autoregressively populated by the predicted tokens. The difference in our case is that each resulting "block" is of a different size. We therefore employ right-masking and padding to enable efficient training and causal inference (see **Fig. 8**).

Note that by just masking our *Current State* and projecting from the $T_n$ row of the block during inference time, we are able to fuse with any other non-masked features, while preventing any backflow of information, and retaining the causal aspect of our model.

# G CALCULATING CONNECTIVITY WITH OTHER METHODS

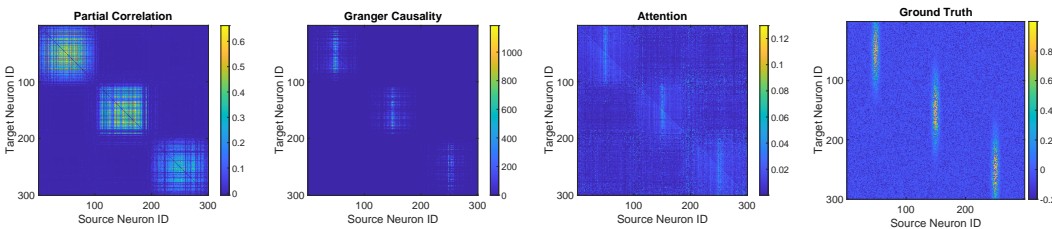

Figure 9: Comparing derived connectivity with other methods. In addition to Pearson correlation, we compared the connectivity from attention to that derived from partial correlation and Granger causality. The square root of the sum of the absolute squares of elements of the attention matrix was 140.4, for Granger causality it was 138.1, and for partial correlation, it was 156.9. These values indicate how each method compares in terms of similarity to the ground-truth connectivity matrix, with lower values suggesting a closer resemblance.

## H   CAPTURING TRIAL-TO-TRIAL VARIABILITY

When investigating the performance of the model as we progressively incorporated each module, we examined F1 scores, and the ability of the model to capture the trial-to-trial variation across the data sets.

To do this, we computed the correlation across trial-averaged spikes for each neuron across groups of four trials. We compared a holdout group to the corresponding set generated by the Neuroformer, and to another corresponding group of trials from the data set. We observed a steady improvement in the distribution of correlations when compared to the ground-truth (**Fig. 10**), with the full model's predictions closely matching those of the ground-truth sets.

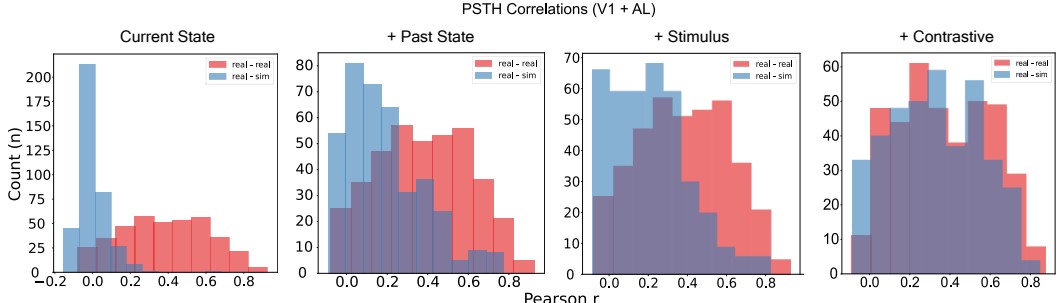

Figure 10: Trial-averaged correlations between groups of 4 trials. The trials generated by the model progressively converged to the correlation distribution of the ground-truth trials as we added components. The model configurations were: *Current State*, *Current State + Past State*, *Current State + Past State + Stimulus*, *Current State + Past State + Contrastive*.

# I    PRECISE BEHAVIOR PREDICTIONS

An advantage of the Neuroformer architecture, is that modalities can be flexibly incorporated as an input or label. In the case of labels, variables can be modeled as both classes as in multi-class logistic regression using a cross-entropy objective (**Fig. 11**), or raw values as a linear regression using mean-squared error (**Fig. 12**). Our model vastly outperformed other baselines, and performed reliably across datasets and objectives, as opposed to other methods, whose performance fluctuated.

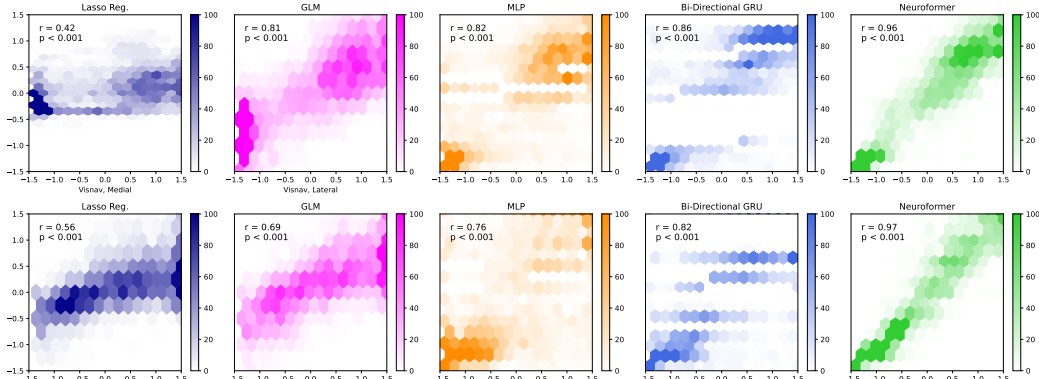

Figure 11: Speed Predictions, Classification. The speed variable was broken into 25 discrete classes and optimized with a multi-class crossentropy loss.

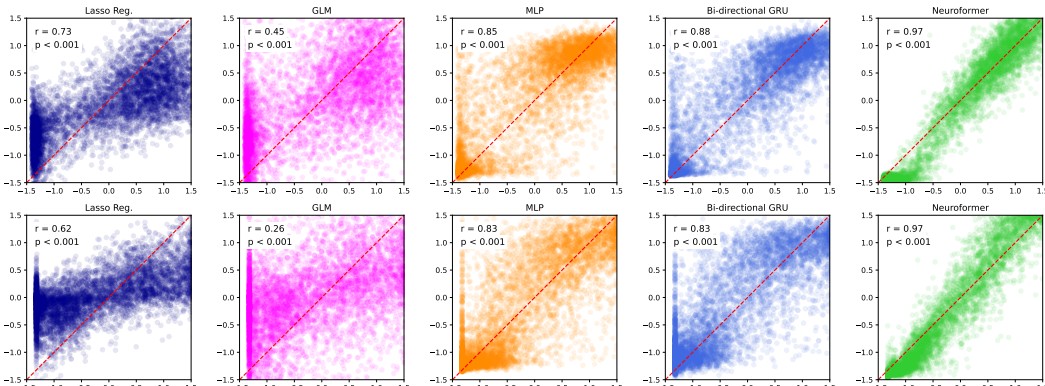

Figure 12: Speed Predictions, Regression. We used MSE loss to optimize the models.

## J  MULTITASK DECODING

In addition to speed, Neuroformer is able to jointly predict many other variables. Here we show that the model could effectively predict the eye position of the mouse - even in the absence of input stimulus. We used a single model checkpoint to decode all three variables. Speed and longitudinal + latitudinal angle for eye position.

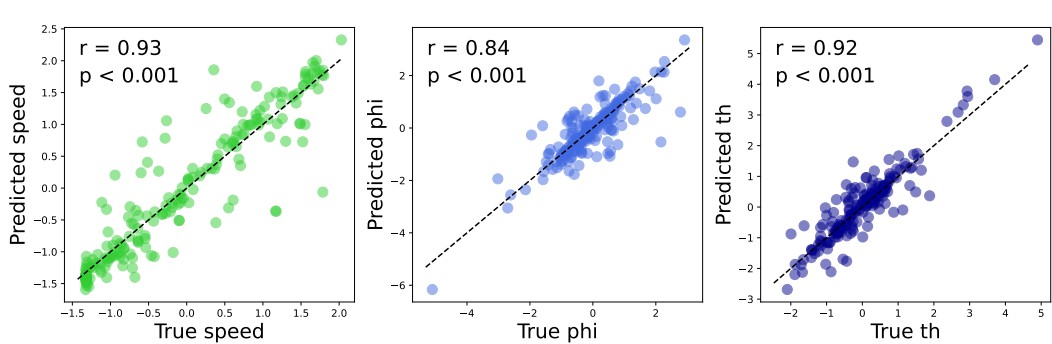

Figure 13: Decoding Speed (m/s) and Eye position phi, th ($^\circ$).

## K  MULTIMODAL LATENT VARIABLES

The contrastive part of Neuroformer, enables us to align any of the corresponding auxilliary modalities with our neural data, including visual features, behavioral features or reward. Here we show the raw features extracted from the contrastive stage that was trained to align speed with neural features.

No dimensionality reduction method (like T-SNE) was used for this visualization. These are the raw features, extracted from our contrastive alignment module, by setting $E_{clip} = 3$.

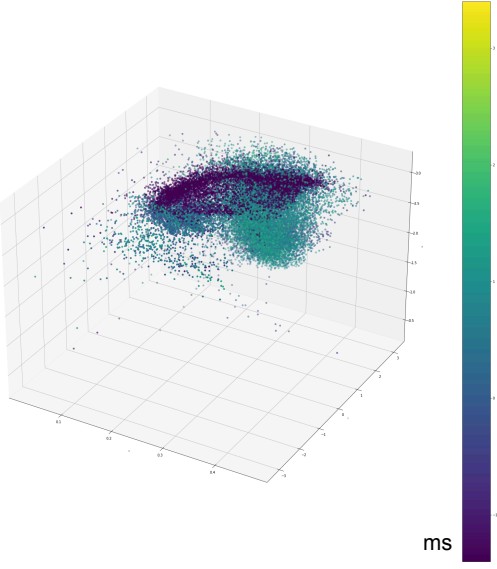

Figure 14: Contrastive Latents: Neural Features Separate According to their Speed.

## L    ANALYSIS OF INTERMEDIATE FEATURES AND PROBING SPEED

### L.1    INTERMEDIATE FEATURES

The features represented in Figure 14 were derived from the contrastive alignment module by setting the $E_{clip}$ to 3, effectively projecting the features to 3 dimensions for alignment during training, as detailed in Hyperparameters 2. Our initial hypothesis posited the need for two dimensions for direction—considering movement occurs on a 2D plane—and one dimension for magnitude. Although we did not expect intermediate representations to fully segregate according to speed without explicit supervision, the process proved beneficial for analytical purposes. As indicated by the toy model experiment in Figure 2, a convergence towards speed-based separation was observed. Additional intermediate representations were extracted from transformer blocks of a model not trained on speed, and a linear regression was performed to map these to speed. These were compared against random arrays of identical shape as a control. supplementary material. As more modalities are fused, the resulting representation becomes more linearly separable in terms of speed.

### L.2    LINEARLY PROBING SPEED

| Feature | $R^2$ |
|---|---|
| Random | 0.0521 |
| Past State | 0.2299 |
| Past State + Stimulus | 0.5341 |
| Past State + Stimulus + Current State | 0.6357 |

Table 3: Linear regression $R^2$ values for different features when probing for speed. Colors coded according to model architecture (Fig. 1.)

# M    CEBRA BEHAVIOR RESULTS

We also attempted to compare Neuroformer behavior predictions with the SOTA in representation learning methods, CEBRA (Schneider et al., 2022). Despite attempting an extensive hyperparameter search accross batch size, temperature, learning rate, offset, and decoding method (KNN and Lasso Reg.) we were unable to get the model to generalize, and typically observed heavy overfitting as seen in **Fig. 15**.

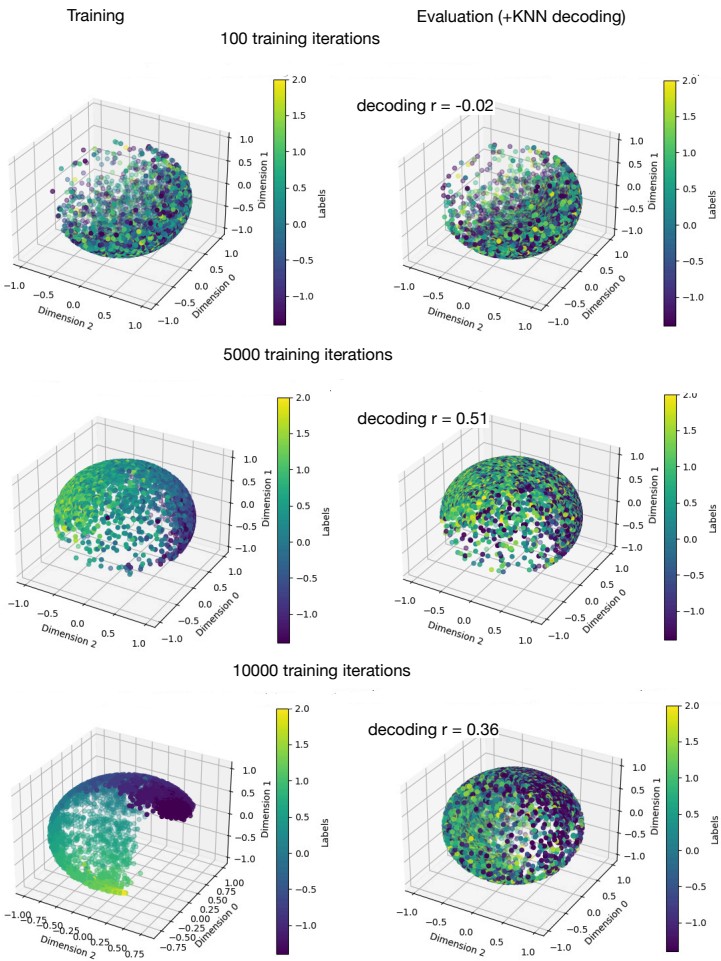

Figure 15: CEBRA decoding. The visualization depicts the latent features produced, colored by speed.

