# OpenReview forum: "Neuroformer: Multimodal and Multitask Generative Pretraining for Brain Data"
_ICLR.cc/2024/Conference — ICLR 2024 poster_

### Official Review · Reviewer_M1PZ · 2023-10-30

**Soundness:** 3 good
**Presentation:** 2 fair
**Contribution:** 2 fair
**Rating:** 6
**Confidence:** 2

**Summary:**

In the paper "Neuroformer: a multimodal, multitask GPT framework for brain data at scale", the authors suggest a transformer-based architecture (Neuroformer) for fitting high-dimensional neural spike train recordings, that can incorporate a CLIP-like contrastive learning objective to use visual stimuli and/or behavioural recordings. The authors argue that Neuroformer can slightly outperform classic GLM models for spike prediction and strongly outperform simpler models for behavioural prediction.

**Strengths:**

The paper is interesting because it applies modern transformer architectures to modeling neural data, and shows competitive results.

**Weaknesses:**

Overall I thought that the paper would perhaps be more suited for a computaional neuroscience journal or for NeurIPS that traditionally has some amount of comp neuro papers. At ICLR, this topic is an outlier, as there is very little (next to none) computaitonal neuroscience there.

Whereas the paper is generally well-written, I found many model details not sufficiently clear (examples below).

**Questions:**

MAJOR COMMENTS

* Section 3: I could not understand the details of the architecture from this description. I could not even understand the basic setup... The text says that each neuron is one token, is that right? So the model is limited by O(1000) neurons, as attention layers scale quadratically with the number of tokens, right? Next, what is one training example: one time bin? For each neuron and each time bin, we have some integer number of spikes. How is this number converted into an embedding vector? What exactly are past states (how many past states) and how are they passed into the model? How does prediction (as in section 4.3.1) work: what is passed as the input instead of neural states?

   Perhaps this is confusing because one would naively think about time-series modeling along the lines of GPT, where time bins (and not neurons) would be tokens. So I think the architecture setup requiers a more detailed explanation.

* Section 4.3.1: the sentence "our model's predictions w[h]ere more closerly corelated with the ground-truth" should contain some quantification, e.g. the fraction of neurons (or what are the dots in figure 3c: are these neurons?) for which Neuroformer outperforms GLM, and also the p-value (0.02) which is currently only mentioned in the figure caption. The evidence here is not very strong, so the authors should not oversell.

* Table 1 is the strongest result in the paper, as Neuroformer strongly outerforms all other models. However, neither the comparison models (Lasso, GLM, etc) nor the prediction task are described in sufficient detail. The task here to predict behaviour from neural responses, but what exactly is "behaviour", what is the input (how many time steps?) and the output (how many time steps) of this prediction problem, etc.? The authors should present their experiment such that it is clear the comparison models in Table 1 are not "strawman" in some sense.


MINOR COMMENTS

* \citep and \citet should be used instead of \cite. Current citation formatting is not following the ICLR template.

* Schneider et al 2022 has been published in Nature.

* page 6: what are N_a and N_z matrices? Unclear.

---

> ### Author Response · Authors · 2023-11-21
>
> Thank you for acknowledging the interesting direction of our work, and the strong results.
>
> **Architecture.** Because multiple authors had similar questions. We have added a global comment explaining the overall workflow, we suggest reading that first.
>
> **Neurons as tokens.** Yes, each Neuron is modeled as a token (**ID**). In a GPT, not all tokens (byte-pair encodings, ~words, typically around ~50,000) are included in every block. Every block (context) consists of the BPE encodings required to form that example/sentence. In our case, we populate the block with the number of Spikes in the current interval. $O(1000)$ would mean 1000 spikes in an interval of 0.05s in our case, which would require a population of tens of thousands of neurons or more (spiking activity is quite sparse). This is one of the advantages of our method. The context is not a sparse matrix of (N_population), but only consists of (N_spikes_per_interval) where typically $ N_{spikesperinterval} << N_{population} $. *Other methods utilizing sparse matrices attempt to compress the representation in order to accommodate larger neuronal populations, but in doing so lose neuronal resolution attention.* This is precisely what we wanted to avoid.
>
> **GLM vs. NF.** We have updated the manuscript to include the actual correlations: the mean correlation for GLM $r=0.232$, Neuroformer $r=0.297$, and for Real-Real $r=0.409$. The correlations were calculated between groups of 4 trials.
>
> **Table 1.** Thank you for recognizing the strong performance of our model on decoding speed. We did not include any past_behavior as input to any of the models, as the models could then simply extrapolate from previous behavior. For all models the output was the speed for the next 0.05s, over the whole of the holdout dataset. For all models bar GRU and Neuroformer, the input was the current state (spikes in the corresponding 0.05 seconds). For the GRU and Neuroformer, which can contextualize over states, we included the past state (0.15s of spikes before current state). We have updated the manuscript to include this information. We actually think figure 5b) is a cooler result, since it shows generatively pretrained models can transfer to decoding and match some of the other methods  with only a fraction of the supervised decoding data.
>
> **N_a and N_z matrices.** Both matrices have shape N x N. N_a denotes the pairwise attention between neurons, and N_z denotes how many times two neurons occur within the same bin. We used N_z to compute the average attention that was attributed between two neurons.

---

> > ### Comment · Reviewer_M1PZ · 2023-11-21
> >
> > Thank you for the response. I also looked at the other reviews by now.
> >
> > I still think the paper is borderline (and arguably not ideally suited for ICLR, but this is not for me to decide), but I appreciate the replies. The authors seem to know what they are doing, and I am not sufficiently familiar with this field to judge on the details. I am going to raise my score to 6 while simultaneously _lowering_ my confidence to 2.

---

> > > ### Author Response · Authors · 2023-11-21
> > >
> > > Thank you for reading our response and raising your score. Just to clarify, the reasons we think the paper is suitable for ICLR are two-fold:
> > >
> > > 1) Encourage the ML community to think about how they can leverage neuronal data within contemporary generatively pretraining models, particularly in areas were other data is sparse, like in continuous control.
> > >
> > > 2) Introduce a mutlimodal, multitask GPT architecture that can simultaneously integrate poth spatial and temporally conditioned signals.

---

### Official Review · Reviewer_dTQU · 2023-10-30

**Soundness:** 2 fair
**Presentation:** 2 fair
**Contribution:** 3 good
**Rating:** 8
**Confidence:** 3

**Summary:**

This paper proposes a multimodal, multitask generative pre trained transformer model called Neuroformer. This model uses an arbitrary number of modalities, such as neural responses,  external stimuli, and behavior, to perform downstream tasks. The authors apply this model to predict simulated neural circuit activities and behavior of a mouse from its neural recordings, where four modalities are used: neural responses, video, speed, and eye position. They also perform ablation studies to explore the impact of each model component.

**Strengths:**

* Propose a multimodal, multitask transformer-based model for neural data modeling.

**Weaknesses:**

* In the experiments, the Neuroformer is only evaluated in terms of behavior prediction, which is not enough in evaluating neuroscience tasks. The choices of models (GLM, GRU, Lasso Regression) for comparison are also not convincing to me. One suggestion is to follow the evaluation criteria in [Neural Latents Benchmark](https://eval.ai/web/challenges/challenge-page/1256/overview) and compare the Neuroformer with the top leading models there, such as S5 and LFADS on the multimodal calcium imaging datasets. It is critical to see whether the proposed model is a solid technical innovation with practical influences in neuroscience research.

* As the authors mentioned in Appendix A, the Neuroformer has poor results (Figure 10) in low-dimensional latent space learning. One possible reason is that the transformer-related neural networks are too expressive so that good dynamics in latent space are no longer necessary. But in terms of interpretability, neuroscientists prefer to observe meaningful latent space in many experimental scenarios, which may be more important than a good behavior prediction performance.

**Questions:**

* What kind of modalities are used in section 4.3? Although we may infer these modalities in section 5, it seems there are no clear describing sentences about them in the whole section 4.3.

---

> ### Author Response · Authors · 2023-11-21
>
> Thank you for taking the time to review our work.
>
> **Neural Latents Benchmark.** NLB is a great benchmark, but it consists of electrophysiology (ephys) data, typically derived from multi-electrode arrays, rather than neuronal-resolution optical physiology (ophys) data which is what our model is designed for  (2-photon calcium imaging data). Please also note that these datasets contain 100-200 electrode areas, while our datasets contain ~2000 Neurons. Our model can be adapted with few modifications to consider this data, and we're excited to explore such directions. Nevertheless, the benchmarks we have used our known to perform on-par with such methods. Please see Paragraph 4 of our global comment.
>
> **Representation Learning Results.** There has been an important misconception here that we want to clarify. Our comment was not intended to mean that our model does poorly at representation learning. We have included some visualizations in the appendix (J, Multimodal Latent Variable) to show the low-dimensional latents the contrastive objective of our model learns. Rather, we wanted to state that some prominent representation learning methods fared poorly on decoding speed from the large-scale calcium imaging data (see K, CEBRA Behavior Results.)
>
> Other than all the methods we tried, including the aforementioned one, the only other prominent method is RADICaL **[1]**, which is an AutoLFADS type method adapted for calcium imaging data, and there is currently no openly available code implementation for the model.
>
> **Modalities.** The modalities used are neural data, video, eye position (latitudinal and longitudinal angle), and speed. The model can additionally incorporate an arbitrary number of modalities or prediction tasks, and can be adapted to do so by just adding a few lines to a config file.
>
> **[1]** Zhu et al., A deep learning framework for inference of single-trial neural population activity from calcium imaging with sub-frame temporal resolution. (2021)  bioRxiv 2021.11.21.469441;

---

> > ### Comment · Reviewer_dTQU · 2023-11-21
> >
> > I thank the authors for the additional explanation and clarification about baseline models. While I still have some concerns regarding the potential efficacy of incorporating evaluation metrics outlined in [Neural Latents Benchmark](https://eval.ai/web/challenges/challenge-page/1256/overview), I acknowledge the authors for their efforts in introducing a multimodal, multitask GPT architecture for extremely high-dimensional neural data with complex experimental settings. This approach contributes to the neuroscience community, particularly in the current "LLM" era. So, I changed the score from 5 to 8.

---

> > > ### Author Response · Authors · 2023-11-21
> > > **Thank you for your considerate response.**
> > >
> > > We appreciate your response, and it reflects our own perspective about our efforts. Our approach brings the fields of Neuroscience and AI closer. We think introducing the "LLM perspective for Neuroscience" could open up interesting new lines of work, for both the aforementioned fields.
> > >
> > > There is a vast amount of untapped Neuroscience data that approaches like ours can leverage. They key difficulty in incorporating them within the same context is their diversity, hence the flexible architecture we have proposed.

---

### Official Review · Reviewer_pMwz · 2023-10-30

**Soundness:** 4 excellent
**Presentation:** 3 good
**Contribution:** 3 good
**Rating:** 6
**Confidence:** 4

**Summary:**

The paper presents a multi-modal Transformer-based pretraining paradigm for learning joint representations of neural activity, behavior and sensory inputs. It mainly follows recent work in vision-language modeling and adapts these approaches to the neuroscience setting. The paper shows that (a) the attention maps can reveal the circuit structure in a simulated toy dataset with a few hub neurons, (b) it can predict neural activity based on past activity and stimulus slightly better than a generalized linear model, and (c) it can decode running speed of a mouse from neural activity in a few-shot manner.

**Strengths:**

+ Promising self-supervised learning paradigm for large-scale, multi-modal neuroscience data
 + Nice set of experiments from simple toy model with known ground truth to real, large-scale data
 + Overall well-written and mostly easy to follow (with some exceptions)

**Weaknesses:**

1. Weak baselines in Figs. 2+3
 1. Small effect in Fig. 3
 1. Architecture (especially decoder) not entirely clear from paper

**Questions:**

Overall I think this is a super interesting and potentially very useful paper, albeit with some weaknesses, which I will detail below. If the authors can address these points in their response, I am happy to reconsider my score.

While I appreciate the experiments, I believe the authors could do a better job at demonstrating that their approach actually works well.

### Fig. 2

It is well known that inferring connectivity from correlations is not a good idea. The most straightforward way of getting closer to connectivity (albeit with a number of limitations and caveats as well) is using partial correlations instead of Pearson correlations. I would predict that the partial correlation matrix would correspond much more closely to the ground truth than the Pearson correlation matrix shown in Fig. 2d. Does your approach perform on par with it or even better?

### Fig. 3

I am somewhat underwhelmed by the result in Fig. 3: All this effort only to improve the correlation between prediction and ground truth by 2%? It may be significant, but the effect size is tiny. Papers on predicting activity from visual stimuli typically show quite more substantial improvements of neural nets over GLMs (e.g. McIntosh, NeurIPS 2016, Klindt et al., NeurIPS 2017, ...). I would like to see some stronger baselines here. The baseline model form the recent Sensorium competition (https://www.sensorium-competition.net) would be a fairly straightforward starting point.

A second point on this figure: I am unsure how to interpret the attention maps in panel d). What exactly do they show? The word "neuron" seems overloaded here. Since the Transformer does not have a token for each neuron (or did I misunderstand something here? According to p.3 bottom the current state does not have a neuron dimension, only batch, time and embedding), I don't quite understand, what, e.g., "Neuron 1, Layer 0, Head 0" means. If it refers to neurons in the brain, please explain how and explain what we don't see localized receptive fields as we would expect from V1. If it refers to something else, please explain what and what we see.


### Architecture not clear

I had a hard time following the description of the method on pages 3+4. Some examples:

 1. This sentence on p.3 could be unpacked: "At each step, a learnable look-up table projects each neuron spike contained in our Current and Past States onto an embedding space E, resulting in vectors (T_c,E) and (T_p,E), where Tc,Tp are the corresponding state’s sequence length (number of spikes + padding)." <-- What does "vectors (T_c,E)" mean? In particular, what is the meaning of the parenthesis around T_c,E? T_c appears to be the sequence length, i.e. scalar. Does it mean you literally concatenate a scalar that indicates the length with a vector E? If so, why? If not, what is happening here?

 1. It is not clear to me what the decoder outputs. From the text and figures I'm guessing it's some form of sparse representation of the spiking in the future, where ID is the row (column) and dt the column (row) of a non-zero entry in a binary matrix (size: #neurons x #timesteps) that contains the spikes. However, I am unsure about this interpretation and cannot map it onto the losses in Eqs. 3+4. Part of the problem might be that p_i and p_dt are not defined and I am not sure how to interpret the cross-entropies. Also, I don't understand why there are two losses. Why not simply output a vector of zeros and ones that is the same size as there are neurons? What is the meaning of dt? I thought you're predicting the next frame?

 1. The last paragraph on p.4 is equally opaque to me and I couldn't make sense of it. The sentence with nucleus sampling is unclear and the meaning of sub-intervals is also not clear.

---

> ### Author Response · Authors · 2023-11-21
>
> Thank you for taking the time to review and acknowledge our work. It makes us happy to see that people understand the ambitious direction we’re pointing towards.
>
> **Architecture.** We have provided a global comment to all reviewers regarding the architecture. We suggest reading that first.
>
> `T_c` and `T_p` are the corresponding current and past states' block size. They are populated with the spikes in those context windows which are made up of two representations: **ID** (the spatial location, or neuron in question) and **DT** (the temporal location, or time at which the spike occurred). To learn DTs, we break each current window into smaller sub-intervals (these can be arbitrarily small, we typically choose the temporal resolution of our measurements as the DT). The IDs and DTs are learnt as tokens, in the same way as words are in a normal GPT model. These tokens have an embedding shape (E). I.e, we learn an embedding of size E for each ID and DT.
> Therefore, the output at each step of the model is not a sparse representation. Rather, it is spikes within that interval. Each individual spike is fed back into the model, and the model autoregressively generates spikes in this way, just like it would do the same using words in the normal GPT setting.
> $p_i$ and $p_dt$ are the probabilities over the corresponding ID and DT tokens for each generation. We sample from these to generate the spikes.
>
> **Nucleus sampling.** restricts the tokens we sample from according to a specified cumulative distribution. I.e, if we choose 0.95, any token that forms part for the cumulative distribution beyond 0.95, is not considered for that timestep.
>
> **Fig 2**. While in figure 2 we weren’t necessarily trying to claim that attention could outperform all other methods, we agree that it would be interesting to compare attention with other methods as well. We have conducted a more thorough comparison using partial correlations, and granger causality. To compute the similarity of the ground-truth connectivity to the different methods, we computed the average sum of squared differences between the matrices (Frobenius norm). The attention matrix was more closely related to the ground truth connectivity matrix. Please see Appendix F for the visualized results.
>
> | Metric              | Value  ↓ (lower is better)  |
> |---------------------|----------|
> | Attention           | 140.4    |
> | Granger Causality   | 138.1    |
> | Partial Correlations| 156.9    |
>
>
> **Fig 3.** The improvement in predictive capability is bounded by the variability between the actual ground-truth responses. In the quoted paper, McIntosh, NeurIPS 2016, Figure 2, the upper-bound is quoted as ~0.8, and this can depend on the way pearson correlation is calculated, i.e bin size. In the same figure, the GLM results look quite low, while the GLM we fitted approaches the ground-truth variability very well. For example, in this [1]  they found that neurons which were < 50 microns from each other and synaptically connected typically had correlation values < 0.4, and that matches other studies of correlation at longer length scales, both within and across cortical areas [2]. Trial-to-trial correlations are typically low in mouse L2/3 neurons because the neurons have low firing rates (<< 1 Hz spontaneous and ~ 5 Hz evoked; e.g., Fig. 8 in Niell & Stryker 2008 J Neurosci). In supplementary materials, “Capturing Trial-To-Trial Variability,” we have included a comparison between sets of 4 ground-truth and predicted trials.
>
> **Sensorium models** are not directly compatible with the data that we used to train Neuroformer, for two reasons. 1) Until 2022, models were trained on images and not video. 2) The baseline model for 2023, while compatible with video, requires precise pupil centers/directions to train the shifter network, which we do not have. Although we have tried in the last days to satisfy this request, we were unable to get well-performing models to compare with. While we still believe that the GLM we used is a strong baseline, particularly when considering the heavy inductive biases used to get good performance, we agree that more comparisons would be desirable.
>
> **Attention.** These attention maps exhibit diversity. Some of the maps seem to highlight large, moving objects, others seem to highlight other mice in the field-of-view, and still other maps seem to be specific for small features, like a mouse’s tail. These are uncharted territory for neuroscience, and we're excited to explore the potential of this method in future work. To reiterate, the attentions shown in the figure are per-token, i.e for an individual spike (neuron), which is one of the advantages of the method, in that it represents neuronal populations at the spike level.
>
> [1] Ko, H., Hofer, S., Pichler, B. et al. Functional specificity of local synaptic connections in neocortical networks. (2011) Nature
>
> [2] Yu et al., Mesoscale correlation structure with single cell resolution during visual coding. (2019)

---

### Official Review · Reviewer_HBVf · 2023-10-31

**Soundness:** 2 fair
**Presentation:** 2 fair
**Contribution:** 2 fair
**Rating:** 5
**Confidence:** 3

**Summary:**

This manuscript introduces a multi-modal, multi-task generative pretrained transformer as a tool for analyzing the increasing volume of data generated by large-scale experiments in system neuroscience. The goal of this tool is to create a better neural spiking model while taking external variables into account.  In particular, they applied the Perceive IO architecture (Jaegle et al. 2021) to the neural domain and modified it accordingly. During the process of decoding, they also developed feature backbones, which enabled the specialized architecture to track the activity of individual neurons. Their loss function has a component that deals with alignment as well as one that deals with spike creation. The alignment component explicitly enforces representational commonalities among biologically significant features. The causal spike modeling is used by the spike generation component to do an autoregressive decoding of brain spikes. They used a simulated dataset in addition to two different two-photon calcium imaging datasets in order to verify the accuracy of this neuroformer. They demonstrated, with the simulated dataset, that the neuroformer is capable of recovering the hub-neuron structure that is comparable to the ground truth. Using the calcium imaging dataset, they were able to demonstrate that the suggested neuroformer performed better than GLM when it came to creating neuronal spikes. They also showed that a pretrained neuroformer has more accurate predictive features of mouse behaviors than baselines like Lasso regression, GLM, MLP, and GRU. In addition to this, they presented the results of an ablation investigation, which demonstrated that each module contributes progressively to predicting eye position.

**Strengths:**

The loss function combines multi-task representation loss with two losses relevant to generating neural spikes. To the best of my knowledge, this particular application of multi-task learning to modeling neural activity is new.

The feature backbones in the Neurofomer are able to dissect single neurons. A common limitation of previous machine learning approaches to modeling population activity is that they lose single neurons. This feature seems to circumvent such a limitation.

**Weaknesses:**

Lack of comparison with strong baselines is my main concern for this submission. Prior to this paper, there were a couple notable publications that leveraged the transformer architecture to generate neural spikes. Albeit those most well-known ones are single modality only, it is still worth a comparison in terms of neural modeling. This work only showed its comparison with simple baselines (MLP, GRU) when it compared the quality of neural spike generation. If this neuroformer does not perform as well as other transformer-based architectures, I would hope the authors may include more elaborated discussion on whether the appeal of cross-modality representation outweighs its limited performance.

1) Liu 2022 Seeing the forest and the tree: Building representations of both individual and collective dynamics with transformers

2) J. Ye and C. Pandarinath, “Representation learning for neural population activity with Neural Data Transformers,” Neurons, Behavior, Data analysis, and Theory, Aug. 2021

The F1 scores in Figure 5 are rather low at their absolute values. It would be helpful if the authors put the F1 score in perspective (why is this F1 score indicating good performance?). Is it possible for the authors to comment on the dip of the F1 score after adding the video modality in the Visnav, Lateral dataset?

The correlation difference in Fig 3C is also low in comparison with GLM. Such a comparison will be a lot stronger if it is versus another transformer architecture or more elaborate architecture that is capable of expressing neural activity fully.

Speed is misspelled as “spped”

**Questions:**

Does this architecture outperform any of those previous approaches in terms of generating realistic neural spikes? Is it possible to show the performance of the neuroformer on the Neural Latent Benchmark?



What is the range of T_p in those calcium imaging datasets? Is it possible to demonstrate long-term inference with a neuroformer?



In Fig. 3d, the attention maps seem to suggest that the intermediate blocks of the neuroformer contain interesting features. It is common for the community that pretrains transformers for time series (like HuBert or Whisper for speech) to use features from intermediate blocks for decoding. Is it possible to show how well these intermediate blocks can be used to predict behavior?



Minor question: I could guess that the red dot is the model being used to generate b) or d). It would be helpful if the authors could clarify this in the figure caption.

---

> ### Author Response · Authors · 2023-11-21
>
> Thank you for the constructive feedback. We were happy to see some of our technical contributions recognized, alas, this seems not to be reflected in the final evaluation score. We hope to alleviate your concerns regarding additional benchmarks. Please also see Paragraph 4 of our global comments.
>
> **F1 scores.** In study **[1]**, neurons within 50 microns and synaptically connected showed correlations below 0.4, consistent with other research **[2]**. Low trial-to-trial correlations in mouse L2/3 neurons are due to their low firing rates, as detailed in Niell & Stryker 2008 J Neurosci. In supplementary materials G, we include a comparison of four sets of ground-truth and predicted trials, showing that our model’s predictions approach the ground-truth variability in responses. Furthermore, please note that in the visnav datasets, each prediction is conditioned on never-before-seen stimulus, as the mouse is freely navigating within a virtual environment setup (please see Figure 7 for experimental setup).
>
> **[1]** Ko, H., Hofer, S., Pichler, B. et al. Functional specificity of local synaptic connections in neocortical networks. (2011) Nature
>
> **[2]** Yu et al., Mesoscale correlation structure with single cell resolution during visual coding. (2019)
>
> **Benchmarks.** Please see our comment (Paragraph 4). The benchmarks we used for comparison are common, and quite competitive even in the NLB settings, where they reach performance close to that of the most performant models on the benchmark, like AutoLFADS. In particular, we tried our best to make a performant GRU by tuning hyperparameters, using tapering, and bidirectionality (for more details see Appendix D).
>
> **Representation Learning for Behavior.** This is an interesting thought. Following your request we have extracted the representations learnt by our contrastive module and provide our results in Figure 14 (Appendix J). The neural features separate according to speed. Please note that no dimensionality reduction techniques were used, these are the raw features from our model.
>
> **Long-Term Inference.** Yes, Neuroformer is able to simulate neuronal responses over very large time horizons. The provided correlation results, and generated spike trains in figure 3b), are all generated using no ground-truth data, over 96 seconds. (w_p) the temporal window for spikes provided to (T_p), is only 0.15s (provided in Appendix C), and other than the first step in the generation of the 96 second trial, all subsequent steps are conditioned on the spikes previously generated. Appendix G shows that these simulations capture the variability of the ground-truth responses.

---

> > ### Author Response · Authors · 2023-11-22
> >
> > We'd like to inform you that other reviewers have updated their assessments. We're thankful for your feedback and are available to address any further questions promptly.
> >
> > If there are no further inquiries, we ask that you consider updating your rating to acceptance, aligning with the consensus of the other reviewers.

---

> > > ### Comment · Area_Chair_wEP3 · 2023-11-22
> > > **Please use the rebuttal period for providing factual clarifications**
> > >
> > > Dear authors,
> > >
> > > the purpose of the rebuttal phase is to provide factual clarifications as input for the reviewers and AC, both so that they can make suggestions for improvement, and that they can discuss the merits and limitations of a paper in order to come to decisions about acceptance. There is absolutely no need for a 'consensus' in this process, and neither are final decisions made based on the numerical scores.
> > >
> > > Best,
> > >
> > > Your AC

---

> ### Comment · Reviewer_HBVf · 2023-11-22
> **Thank you for your rebuttal**
>
> I appreciate the authors' responses to my questions. I read through the author's responses to all reviewers. This paper, in my opinion, is, at best a borderline case.
>
> As the primary demonstration of this work pertains to the generation of neural spikes, I proposed a comparison with alternative transformer-based architectures without specifically referencing the Neural Latent Benchmark. I recognize that the author may find it difficult to modify those alternative architectures to fit their data within the time constraint of the rebuttal period. I then proposed a discussion of cross-modality versus single-modality. In the absence of both comparisons, it is difficult for a system neuroscientist to migrate their modeling to cross-modality, given the potential difficulty of training (again, this is merely my conjecture; the author's response implied that they are unable to get the architecture to function on the neural latent benchmark). Its potential impediments could impede its applicability within the system neuroscience community.
>
> A well-fitted GRU should not replace the comparison of alternative transformer architectures. The GRU fitted to the author's datasets would differ from the GRU fitted to NLB. To claim that comparing to GRU is sufficient is akin to comparing pears and oranges. Critically, I asked about the comparison with alternative transformer architectures due to the potential for the attention head to contain interesting information about stimulus encoding.
>
> I also find the speed decoding shown in Fig 14 not convincing. There is clear overlap between points of different colors (or maybe it is easier to see separation if the opacity level is less than 1?). I’d also appreciate some clarification of having the dimension of neural features as 3. I could not find such architecture details in Appendix C.
>
> I echo with the Reviewer pMwz that the architecture details are unclear.
>
> I will raise the score to 5, and this is my final score for this submission.
>
> Minor
>
> I also find the updated paper a bit sloppy in its presentation. 3 subfigures in Fig 2 are missing. 9/10 subfigures in Fig 12 are missing.

---

> > ### Author Response · Authors · 2023-11-23
> >
> > Thank you for taking the time to respond, read our discussion with other reviewers, and also for reviewing the changes we have made in the manuscript. We hope you have had time to also review the architecture animation we shared in the supplementary materials in hopes that it can clarify some questions about the architecture. We are happy to help clarify any more details.
> >
> > **Uni/Multi-modality.** In terms of the discussion between unimodal and multimodal models, providing specific explanations for the performance of models trained on different modalities is a complicated question that is difficult to answer within the scope of our work. The lateral area we imaged is rich in visual features, more so than the medial area, which is thought to contain more behavioral information. In Figure 6, notice how upon adding the eye position, the lateral performance increases again. One explanation could be that the eye position information is helping the model prevent overfitting on the video by indicating the features the mouse is actually looking at. This is quite speculative, but this is the type of analysis we are hoping Neuroformer encourages the community to undertake - a data-driven analysis across modalities and tasks of increasing complexity. Here, our attempts were to present the model and a case study on its use on both our simulated and real data. Perhaps in future work we could apply the model to NLB (and in hindsight, focussing on a benchmark would have made our lives easier in the review process). Having said that, our hope is that the model is used for analysis of diverse, previously untapped neuroscience datasets, in order to facilitate neuroscientific discovery and improve current ML models. Demonstrating utility on 4 datasets with different structure and modalities is good evidence that our approach has the potential to do that.
> >
> > **GRU** In terms of the GRU, it wasn't our intention to suggest that the GRUs we used were the same as the ones in the NLB paper, but rather that the relative performance between GRU and other methods in that paper could suggest that in general the GRU is quite a strong baseline to compare with. In the same way, using NDT or similar methods adapted from NLB would require changing the models to the point were we would be creating a new model - particularly since they use much stronger inductive biases than a GRU or our model.
> >
> > **Intermediate features** The features in Fig. 14 came from the contrastive alignment module. For the experiments to extract the 3D latents, we set the $E_{clip}$ (see Hyperparameters, Appendix C) to 3, projecting all the features to 3 dimensions for alignment during training (our hypothesis was 2 dimensions for direction, since the mouse is moving on a 2D plane, and 1 dimension for magnitude). We wouldn't expect intermediate representations to completely separate into speed without any explicit supervision. Your suggestion is nevertheless a good analysis case. Figure 2 (toy model experiment) is pointing towards the same direction. We have further extracted intermediate representations from the transformer blocks of a model not trained on speed, and fit a linear regression line mapping them to speed. We also use random arrays of the same shape, as a control. Note that the mapping is from a 256-dimensional embedding. We have included the results as a jupyter notebook (`neuroformer_extract_latents.ipynb`) in the supplement. Interestingly, *as we fuse more modalities, the representation becomes more linearly separable in terms of speed.*
> >
> >   ###    Linearly Probing Speed
> > | Feature                     | R2               |
> > |-----------------------------|------------------|
> > | random                      |  0.0521 |
> > | past state                     | 0.2299 |
> > | past state + stimulus          | 0.5341 |
> > | past state + stimulus + current state  | 0.6357 |
> >
> > **Manuscript** We have improved the Multimodal Latent figure following your feedback. As for the missing panels, on our end, we can see all of them, but perhaps this issue has arisen because we are embedding heavy vector figures. We will investigate further and potentially rasterize some of those to eliminate this issue.
> >
> > Thanks again for your valuable feedback and your thorough evaluation of our work.

---

### Author Response · Authors · 2023-11-21
**Thank you for your time and constructive comments.**

Thank you to all the reviewers for their time and constructive comments. We were happy to see that they have recognized some of the novelties of our work. Particularly about incorporating multi-task generative representation loss for neural data model’s ability to dissect neural representations at the neuronal level, which is what enables our first toy connectivity experiment and opens up interesting future applications.

There were some questions about the architecture which we thought we needed to address. In a typical GPT model tokens take the forms of words. These words are embedded with positional encodings since the sequence of tokens (T=block size, E=embedding size) do not contain position information. In the Neuroformer, the equivalent of a vocabulary is the number of neurons in the population (**IDs**. The big difference between spikes and words though, is that *while the relative position of words is self-evident, the relative positions of spikes is not, and need to also be predicted.* Therefore, we learn a second vocabulary, of (**DTs**), which specify when the neuron fired. At each step, **Neuroformer outputs the ID of the next spike, and the DT in which it has fired**. This DT is added to the ID as a temporal embedding to indicate the IDs position in time (the equivalent of a positional encoding in a normal sentence.) Notice that by doing this, we can auto-regressively predict simultaneous spikes (by predicting the same dt for them) which solves the major issue in using GPT for neuronal population generation. Note that this architecture can not only be used for neural data, but also to on any spatio-temporal signals (such as audio).

In terms of other baselines, we have devoted extensive time to figuring out the fairest baselines to compare our method. The GLM we used was hand-designed for the specific dataset we used. We would suggest that the fact that Neuroformer outperformed this gold-standard model while including minimal inductive biases, which allow it to incorporate further modalities and tasks, is a positive indication in proving GPTs can work for neuronal data. This was not a priori obvious.

We have attempted to adapt representation learning approaches to our calcium imaging datasets, but despite devoting a good amount of time, we were unable to get them to work. Please see Appendix K. Works such as AutoLFADS and NDT are adapted for high-frequency electrophysiology data instead. To this day, work on behavioral decoding from calcium imaging remains sparse and underdeveloped. Nevertheless, we’d like to mention that recent work has shown that **GRUs in particular come very close to the performance of AutoLFADS on the NLB benchmark**, which indicates that the GRU is quite a strong baseline to compare with our method. We therefore believe this shouldn’t form grounds for rejection of our work.  [1] Table 3. NLB-Maze: GRU - 0.8887, AutoLFADS - 0.9062.

We encourage the reviewers to evaluate our work for what it is rather than what it is not. A framework that introduces generative pretraining for multi-task, multi-modal systems neuroscience data. For a further discussion of our work’s limitations please refer to Appendix A. We’re thankful to the reviewers for taking the time to read through our work and are available for any further clarifications.

**[1]** Azabou, M., Arora, V., Ganesh, V., Mao, X., Nachimuthu, S., Mendelson, M. J., Richards, B., Perich, M. G., Lajoie, G., Dyer, E. L. A Unified, Scalable Framework for Neural Population Decoding. (2023) arXiv:2310.16046 [cs.LG].

---

> ### Author Response · Authors · 2023-11-21
> **Architecture animation, attention video, code.**
>
> We are sharing an anonymous link with an animation showing how spikes are auto-regressively generated, a video showing average attention per-time step from the point-of-view of the mouse, and our current codebase.
>
> Note: To avoid any issues we have uploaded the aforementioned materials in the supplementary section.

---

### Meta-Review · Area_Chair_wEP3 · 2023-12-05

**Metareview:**

The paper introduces a multi-modal, multi-task generative pretrained transformer to analyse large-scale neural recordings in system neuroscience. The authors make careful choices to adapt the an existing architecture (Perceiver IO) to this domain.  The method is demonstrated on simulated data and  two different two-photon calcium imaging datasets.

**Justification For Why Not Higher Score:**

It is an ok paper, but not outstanding.

**Justification For Why Not Lower Score:**

Reviewer recommended acceptance, and I see no reason to overrule them.

---

### Decision · Program_Chairs · 2024-01-16

Accept (poster)